

# Conflict resolution in socially housed Sumatran orangutans (*Pongo abelii*)

Kathrin S. Kopp and  Katja Liebal

Department of Education and Psychology; Comparative Developmental Psychology, Freie Universität Berlin, Berlin, Germany

Corresponding author
Kathrin S. Kopp,
kathrin.kopp@fu-berlin.de

## ABSTRACT

**Background**.  Peaceful conflict resolution strategies have been identified as effective mechanisms for minimising the potential costs of group life in many gregarious species, especially in primates. The knowledge of conflict-management in orangutans, though, is still extremely limited. Given their semi-solitary lives in the wild, there seems to be barely a need for orangutans to apply conflict management strategies other than avoidance. However, because of the rapid loss of orangutan habitat due to deforestation, opportunities to prevent conflicts by dispersion are shrinking. Additionally, more and more orangutans are brought into rehabilitation centres where they are bound to live in close contact with conspecifics. This raises the questions of whether and how orangutans are able to cope with conflicts, which are inevitably connected with group life.

**Methods**. Observational zoo-studies provide a valuable method to investigate such potential: in zoos, orangutans usually live in permanent groups and face the challenges of group life every day. Therefore, we observed a group of six socially-housed Sumatran orangutans at the Dortmund Zoo, Germany, both in their spacious outdoor enclosure in the summer and in the less spacious indoor enclosure in the winter. During 157.5 h of observation, we collected data on aggressive interactions, third-party interventions and post-conflict affiliations. We applied the post-conflict/matched-control observation (PC/MC) and the time rule method to investigate the occurrence of reconciliation and post-conflict third-party affiliations.

**Results**. We recorded a total of 114 aggressive interactions (including conflicts in the context of weaning and of male sexual coercion). As expected, we found an increase of both open conflicts and peaceful conflict resolution under less spacious conditions. In accordance with previous reports, we observed interventions by initially uninvolved individuals. Whereas we found no clear evidence for post-conflict third-party affiliations, we were able to demonstrate the occurrence of reconciliation among orangutans.

**Discussion**. Notwithstanding the small sample size and the explorative character of our study, we found evidence that orangutans possess a potential for prosocial conflict resolution. When living in groups and under conditions in which dispersion is no longer an option, orangutans are capable to flexibly apply strategies of conflict resolution to cease open conflicts and to repair the potential social damage of aggressive interactions. These strategies are similar to those of other great apes.

[1]Annotation: The present article, including most of the figures and tables, is largely based on Chapter 2.3 of the dissertation of the first author (*Kopp, 2017*).

# INTRODUCTION[1]

When *de Waal & van Roosmalen (1979)* published their seminal study on reconciliation and consolation in chimpanzees (*Pan troglodytes*), they initiated a shift in perspective in the research on conflicts: from focussing primarily on aggression to investigating conflict situations and how they are managed (*Aureli & de Waal, 2000b*). Conflicts of interest regarding, limited resources such as food or mating partners (*Janson, 1988*; *Robbins, 2008*) may potentially result in direct costs including loss of resources or injuries, but also in indirect costs by jeopardising valuable social relationships (*Aureli & de Waal, 2000b*; *Kummer, 1978*). Given that long-term relationships are crucial for the exchange of benefits within primate groups (*Furuichi & Ihobe, 1994*; *Langergraber, Mitani & Vigilant, 2009*; *Mitani, 2009*; *Seyfarth & Cheney, 2012*), their endangerment might be even more disadvantageous to an individual than the immediate costs. Conflicts may also destabilise the whole group, when social networks are affected (*Flack et al., 2006*).

To minimise the costs, a variety of conflict management mechanisms have evolved across group-living species (*Aureli & de Waal, 2000a*). These mechanisms include prosocial conflict resolution, such as interventions by third parties, and post-conflict affiliations either between the opponents (*reconciliation*) or between an opponent and a third party (*post-conflict third-party affiliation or PCTA*) (*Judge, 2003*). Prosocial conflict resolution may have several, not necessarily mutually exclusive, functions, e.g., protecting valuable partners and stabilising or restoring valuable relationships (e.g., *Aureli, Cords & van Schaik, 2002*; *Cords & Killen, 1998*; *Kutsukake & Castles, 2004*; *Palagi, Paoli & Borgognini Tarli, 2004*; *Silk, 2002*; *Wittig et al., 2007*), reducing conflict-related anxiety in the involved individuals (e.g., *de Waal & Aureli, 1997*; *Fraser, Stahl & Aureli, 2008*; *Romero & de Waal, 2010*) or maintaining group cohesion (*Flack et al., 2006*).

Much of our knowledge comes from studies on the African great ape species, especially chimpanzees (*Pan troglodytes*) and bonobos (*Pan paniscus*), who regularly engage in third-party intervention, reconciliation and PCTA (e.g., *Fraser & Aureli, 2008*; *Palagi & Norscia, 2013*; *Palagi, Paoli & Borgognini Tarli, 2004*; *Rudolf von Rohr et al., 2012*). Their social systems are characterised by a large community size, multimale/multifemale groups and a high degree of fission–fusion dynamics demonstrating social flexibility (*Aureli et al., 2008*). Social bonds, reciprocity and cooperation play a central role in both species (*Boesch, 1994*; *Furuichi & Ihobe, 1994*; *Jaeggi et al., 2013*; *Langergraber, Mitani & Vigilant, 2009*; *Mitani, 2009*). Hence, an important function of conflict resolution in these species most likely consists in stabilising and restoring valuable relationships (*de Waal & Aureli, 1997*). Gorillas (*Gorilla* spp.) live in smaller, more stable and cohesive, predominantly one-male harem groups (*Parnell, 2002*; *Robbins, 1995*); and male interventions in conflicts among females as well as affiliative post-conflict contacts are common (*Cordoni, Palagi & Tarli, 2006*; *Scott & Lockard, 2007*; *Watts, 1995a*; *Watts, 1995b*; *Watts, 1997*).

In contrast, our knowledge about conflict resolution in the Asian great apes, the various species of orangutans (*Pongo* spp.), is very limited. Although the social organisation of orangutans has been characterised as an individual-based fission–fusion society (*van Schaik, 1999*), orangutans are largely known as semi-solitarily living species with a mean

party size of less than two individuals (*Delgado & van Schaik, 2000*). However, sociability varies between the species, with Sumatran orangutans (*Pongo abelii*) being generally more gregarious than Bornean orangutans (*Pongo pygmaeus*), but also within a species or over time (*Delgado & van Schaik, 2000*; *Husson et al., 2009*; *van Schaik, Marshall & Wich, 2009*). For example, in periods of high fruit abundance, orangutans may aggregate in large fruit trees and occasionally form travel bands (*Delgado & van Schaik, 2000*). Adult females and their dependent offspring live in large, overlapping home ranges, which overlap with the larger home range of a resident flanged male (*Singleton et al., 2009*; *van Schaik, 1999*). In contrast to other great apes, orangutan females are the philopatric sex (*Arora et al., 2012*; *van Noordwijk et al., 2012*), i.e., females tend to stay in their natal area, whereas mature males disperse (*Mitra Setia et al., 2009*). Maternally related females with dependent offspring associate with each other and tolerate or even enable social play among their offspring (*van Noordwijk et al., 2012*). At some sites, they are unusually tolerant among each other, feed in close proximity and even share food (*Singleton & van Schaik, 2002*; *van Schaik, 1999*). Females and males temporarily engage in consortships, characterised by coordinated traveling, cooperative mating and other social interactions (*Utami Atmoko et al., 2009a*), however, forced copulations occur often—especially by unflanged males.

In sum, recent findings suggest that orangutans are not as solitary as suggested and that social relationships might play a more important role, especially among females, than previously assumed, raising the question whether conflict resolution might be present also in orangutans. However, there is almost no evidence that wild orangutans apply conflict management strategies other than avoidance. This strategy is most obvious for sexually mature (flanged) males, but also for adult females. Flanged males are highly intolerant of other flanged males, probably due to high mating competition, and rarely encounter each other, although their large home ranges overlap. If encounters occur, they are inevitably agonistic and often result in injuries or even death (*Utami Atmoko et al., 2009b*). Associated unrelated females show more aggressions among each other than related females. These aggressions usually result in breaking up the association; related females, on the other hand, show more social tolerance (*van Noordwijk et al., 2012*). The only report of a form of prosocial conflict resolution in the wild refers to a case, when a female Bornean orangutan received protection from a flanged male, who intervened actively, though non-aggressively, in a sequence of severe, finally lethal attacks by another female and an unflanged male (*Marzec et al., 2016*).

However, given the rapid loss and fragmentation of orangutan habitat due to deforestation, the opportunities to prevent conflicts by dispersion are shrinking (*Nellemann et al., 2007*). Additionally, as a consequence of habitat loss, poaching and illegal pet trade, a growing number of orangutans are destined to live in rehabilitation centres in close contact with conspecifics, partly under overcrowded conditions (*Russon, 2009*). Therefore, the question as to whether and how orangutans are capable of coping with inter-individual conflicts, which are unavoidably connected with group life, needs to be answered. A better understanding of these capabilities may also improve the welfare of orangutans living in rescue centres or zoos due to adjusted husbandry conditions.

To tackle this problem, observational zoo-studies provide a valuable approach to investigate the conflict resolution potential in question. In modern zoos, orangutans are mostly kept in permanent groups, usually comprising one adult male, several adult females and their offspring. Whereas this group structure accounts for the natural dispersal strategy of male orangutans (*Knott, 2009*), it simultaneously precludes any interactions among flanged males as well as any female partner choice. Furthermore, these groups often comprise unrelated females.

Despite these differences of social life between natural and captive conditions, captive orangutans generally seem to cope surprisingly well with group life, and open conflicts are rare. Zoo-living orangutans engage in social play and other affiliative interactions (*Edwards & Snowdon, 1980*; *Zucker et al., 1986*; *Zucker, Mitchell & Maple, 1978*), they develop social bonds even among not kin-related individuals, and share food selectively with close social partners (*Kopp & Liebal, 2016*). Group-living orangutans tend to communicate over food and share frequently when provided with a monopolisable food source (*Kopp, 2017*; *Kopp & Liebal, 2016*; *Liebal & Rossano, 2017*), which demonstrates their ability to mitigate conflicts by tolerant and prosocial behaviour. Considering these findings on the one hand and the phylogenetic proximity between orangutans and the other great ape species on the other hand, group-living orangutans should also be able to cease conflicts, to reconcile and/or engage in PCTAs with close and/or valuable social partners after open conflicts.

However, to date, conflict resolution among captive orangutans has been neglected by scientific research—apart from very few reports on third-party interventions (*Tajima & Kurotori, 2010*; *Zucker, 1987*). With this observational study, we aim to fill this gap and postulate the following hypothesis:

H: The behavioural repertoire of orangutans includes strategies of conflict resolution that are similar to those of the other great apes. Group-living orangutans use these strategies in order to reduce direct and indirect costs of open conflicts.

We derived the following predictions:

(P1) If there is the opportunity to avoid confrontations, open conflicts should be rare. With decreasing available space, both open conflicts and conflict resolution are expected to increase in frequency.

(P2) Third parties are expected to intervene in aggressive interactions, especially when they are highly intense and/or a valuable social partner is involved.

(P3) Immediately after a conflict, former opponents are expected to engage in affiliative contacts with each other.

(P4) Immediately after a conflict, an increased number of affiliative contacts between former victims and third parties are expected.

Findings meeting these expectations would provide evidence for conflict resolution behaviours in orangutans being similar to those found for other great apes. Therefore, they would further support the more general hypothesis that conflict resolution has evolved in a common ancestor of all extant hominids.

**Table 1  Details of observed individuals: sex, age and kin relation with other group members.** Age categories have been assigned following *van Noordwijk & van Schaik (2005)*. A and B, resp., refer to the subgroups formed during the winter.

| Individual | Sex | Date of birth | Age category | Subgroup | Kinship |
|---|---|---|---|---|---|
| *Walter* (Wa) | Male | 24/04/1989 | Adult | A | Father of Ta and Ei |
| *Toba* (To) | Female | 07/02/1994 | Adult | A | Mother of Ta and Ei |
| *Tao* (Ta) | Female | 18/11/2004 | Semi-dependent immature | A | Daughter of Wa and To |
| *Eirina* (Ei) | Female | 30/12/2007 | Dependent immature | A | Daughter of Wa and To |
| *Suma* (Su) | Female | 14/03/1993 | Adult | B | No kin |
| *Djamuna* (Dm) | Female | 28/05/1999 | Adult | B | No kin |

## MATERIALS AND METHODS

### Study group and housing conditions

Observations were conducted on a group of six Sumatran orangutans at the Dortmund Zoo, Germany, consisting of one adult male and three adult females, one of them with two offspring: one not fully weaned, semi-dependent female and one unweaned, dependent female (Table 1).

In the summer, all individuals were kept together in the large grassy outdoor enclosure (1,515 m$^2$) during day time, featuring trees, bushes, herbs, climbing structures with ropes, a tree hut and a fresh water spring. During the winter, the individuals were kept in two subgroups (Table 1) for zoo management reasons. The subgroups were housed in two adjacent indoor enclosures (65 m$^2$ and 48 m$^2$), each with additional night boxes and alternating temporary access to a third indoor compound (140 m$^2$). Subgroup A consisted of the male *Walter*, the female *Toba* and their two daughters *Tao* and *Eirina*; subgroup B consisted of the two unrelated females *Suma* and *Djamuna*. All indoor enclosures were covered with a steel mesh and equipped with climbing structures and ropes, poking timber and access to water. An additional mesh, which separated the enclosures, provided the opportunity for orangutans to reach through with their hand (adults) or arm (immatures). Both subgroups, therefore, had—and frequently used—the opportunity to interact with each other through the mesh, e.g., by exchanging food or non-food items, sitting in body contact, playing or even mating (see also *Kopp, 2017*, pp. 47, 58; *Kopp & Liebal, 2016*). Aggressive interactions were also possible, including grabbing body parts or pulling their fur, hitting, biting or chasing each other across the mesh.

The main diet consisted of a mixture of vegetables and fruits, supplemented by, leafy branches, yoghurt, cooked eggs and meat and special items for behavioural enrichment. Feeding times did not change depending on the housing conditions.

### Data collection and coding

Group observations were conducted over a total of 30 days during September 2011, February/March 2012 and June/July 2012, with an average observation time of 5.25 h per day. Using a digital camcorder CANON Legria FS200 (Tokyo, Japan), the whole group was continuously video recorded by KK. This produced 157.5 h of footage in total, of which 74.35 h had

been collected outdoors and 83.15 h indoors. The filming sessions were evenly distributed over the daily main activity period in order to cover all situations typically occurring in the group.

Data were coded using Microsoft Excel® 2010 and the coding software INTERACT® Vers. 14.3. We considered all aggressive interactions and interventions by third parties as well as all post-conflict affiliative interactions, the latter occurring both between the former opponents (*reconciliation*) and between the victim and a third party (*post-conflict third-party affiliation, PCTA*). To code aggressive interactions and third-party interventions, we applied *all occurrences sampling* as the sampling rule (*Altmann, 1974*).

An *aggressive interaction* was defined as an interaction between two or more individuals that comprised an initial aggression by an individual and a respective reaction by a target individual. *Aggression* was defined as a directed behaviour of an individual (*aggressor*) towards a group member (*victim*) that resulted in physical harm or signalled the readiness to harm (adapted from *Aureli & de Waal, 2000a*, p. 387). To determine whether an interaction such as wrestling or chasing was playful or aggressive, the reaction of the target individual was taken into account. If the behaviour in question caused distress of the recipient, expressed, for example, by pilo-erection or vocalisations such as *kiss squeaks*, *whimpering* and *screaming*, or avoidance and resistance behaviour, this behaviour was coded as aggressive. With regard to the intensity of an aggression, we distinguished low, medium and high levels of aggression. When an aggression consisted not of a single behaviour, but of a combination of several behavioural elements, its level of intensity was categorised with regard to the most intense constituent. A *reaction to an aggression* could consist of a single behaviour or a combination of behaviours belonging to one of four categories: avoidance, non-aggressive behaviour intended to cease the attack, aggressive behaviour against the initial aggressor, i.e., counter-aggression, or against an uninvolved third party, i.e., redirection (*Aureli & de Waal, 2000a*, p. 387). We defined a *third-party intervention* as the attempt of an initially uninvolved individual (or individuals) to cease an ongoing open conflict through peaceful behaviour, e.g., appeasement behaviour, shielding the victim or separating the opponents, or aggressive behaviour, e.g., attacking the aggressor (for details, see coding scheme, Fig. S1).

To code the victim's affiliative post-conflict contacts, we combined behavioural and focal sampling (*Altmann, 1974*) and applied the post-conflict/matched-control observation method (PC/MC method, *de Waal & Yoshihara, 1983*). Any kind of friendly interactions were coded, such as *contact sitting*, *embracing*, *social play*, *food sharing*, *grooming* or *gently touching*. We also determined who initiated the affiliative contact.

For PC observations, recordings of the first 10 min following an aggressive interaction were coded with a focus on the victim. The victim's first affiliative contacts with the former aggressor, on the one hand, and with an uninvolved individual, on the other, were coded. To determine the latency of their occurrence with respect to the end of the aggression, the respective 1-minute time interval in the PC period was assigned to each first affiliative contact. In case the conflict had been resumed within the first 3 min of the PC period, we stopped this PC and began a new PC immediately following the end of the resumed aggressive interaction. Each PC was paired with a particular MC. For MC observations, we

examined the video material recorded on the next possible day for 10 min to code the first affiliative contacts of the former victim with the former aggressor and with a third party, respectively. These observations were usually conducted on the very next day at exactly the same time; if this was not possible, we did so within a time slot of maximum ±60 min with regard to the onset of the exactly matched time. If the necessary next-day MC recording was not available, we used time-matched video footage of observations on a previous or following day within a maximum time window of ±seven days. If a conflict had taken place within 10 min prior to the planned MC interval, we chose the interval nearest to the matched time, within a time slot of maximum ±60 min, which followed an interval without conflict of at least 10 min. Applying this method resulted in PC-MC pairs, which were analysed as described in the following section.

## Data analysis

All statistical computations were conducted by using statistics software R vers. 3.4.1 (*R Core Team, 2017*) with additional packages, e.g., car (*Fox & Weisberg, 2011*) and sfsmisc (*Maechler et al., 2016*) (for more details, see below). Statistical significance was assessed at the $\alpha$-level of 0.05.

### *Aggressive interactions and third-party interventions*

In a first step, we calculated the frequencies of aggressive interactions with regard to context, intensity and the identity of the opponents.

To test for an effect of the housing conditions on the occurrence of aggressions, we computed the aggression rates per hour for outdoor (74.35 h) and indoor observation (83.15 h) for each individual. Assuming that the aggression level would rise under less spacious conditions, we conducted an exact one-tailed Wilcoxon signed rank test using the R function wilcox.exact from the R package exactRankTests (*Hothorn & Hornik, 2015*).

We conducted a weighted social network analysis (Fruchterman-Reingold algorithm) in order to visualise behavioural patterns of aggressions within and between individuals separately for the two housing conditions, by using the R packages igraph (*Csardi & Nepusz, 2006*) and tnet (*Opsahl, 2009*).

For possible effects of the housing conditions on the intensity of aggressions and the occurrence of third-party interventions, we conducted Fisher's exact tests that account for a small sample size using R function fisher.test. If possible, we calculated odds ratios. To control for influential individuals, we ran Fisher's exact tests of independence. When no significant correlation between individuals and the distribution for outdoor and indoor enclosure could be detected, we considered both variables as independent, i.e., the effect as being not driven by particular individuals. We also investigated whether the mesh– separating the subgroups indoors and through which many aggressions occurred—might have had an effect on the frequency of aggression. For that purpose, we ran a Fisher's exact test of independence on pooled data (indoor vs. outdoor; separated dyads vs. not separated dyads).

Forced copulations represent a specific class of aggression in orangutans, which is uncommon in other non-human great apes. Therefore, we separated these cases from aggressions in other contexts and considered both subsets separately in the further analyses.

Due to the rare occurrence of intervened conflicts, inferential statistics on an individual level were not possible. Therefore, we applied descriptive statistics. In order to test whether the more limited space would increase the probability of third-party interventions, we conducted a Fisher's exact test on pooled data and calculated the odds ratio.

### Affiliative post-conflict contacts

In a next step, we examined whether and how conflicts altered the subsequent behaviour between the opponents as well as between other group members and the victim (*Veenema, Das & Aureli, 1994*). In particular, we were interested in whether reconciliation and/or post-conflict third-party affiliation occurred. *Reconciliation* was defined here generally as the affiliative reunion of the opponents evoked by the respective conflict. *Post-conflict third-party affiliation (PCTA)* was defined here generally as the first affiliative interaction between a previously uninvolved individual and the former victim evoked by the respective conflict, regardless of the identity of the individual who initiated the contact.

To investigate whether reconciliation or PCTA occurred, we combined two well-established methods as recommended by *Veenema (2000)*: the PC/MC method (*de Waal & Yoshihara, 1983*) and the time rule method (*Aureli, van Schaik & van Hooff, 1989*). Both methods have their advantages: whereas the PC/MC method controls for inter-individual differences by comparing PC and MC for the particular dyads, the time rule method allows for an operational definition of reconciliation by determining a relevant time window following a conflict (*Veenema, 2000*, p. 22).

*PC/MC method.* We differentiated between the following types of PC-MC pairs (*de Waal & Yoshihara, 1983*): a PC-MC pair was *attracted* if the affiliative interaction took place earlier in the PC than in the MC, or only in the PC. If the affiliative contact took place earlier in the MC than in the PC, or only in the MC, then it was categorized as *dispersed*, and as *neutral* if either the affiliative interaction occurred in both observation intervals at the same time, or did not occur at all.

Following *de Waal & Yoshihara (1983)*, a significantly higher proportion of attracted vs. dispersed PC-MC pairs would demonstrate the occurrence of reconciliation or of PCTA in the study group. To test this, we conducted exact Wilcoxon signed rank tests on the proportions for attracted and dispersed pairs for each focal individual (*Fraser & Aureli, 2008*, p. 1116), using the R function wilcox.exact from the R package exactRankTests (*Hothorn & Hornik, 2015*). We included only those individuals who had received aggression at least three times.

*Time rule method.* As an alternative means of determining reconciliation and PCTA, which allows for an operational definition of both concepts (*Aureli, van Schaik & van Hooff, 1989*), we indicated for each PC-MC pair the respective one-minute intervals in which the first affiliative contacts between the opponents or between the victim and a third party occurred. Then we computed the cumulative distributions (i.e., the cumulated relative frequencies) over time for PC and MC. Following *Aureli, van Schaik & van Hooff (1989)*, we applied a two sample Kolmogorov–Smirnov test, using the R function ks.test to investigate whether the distributions differed. In case a significant difference could be

demonstrated, we determined the time interval with the maximum difference between the cumulative relative frequencies of PC and MC to operationally define reconciliation. Following *Aureli, van Schaik & van Hooff* (*1989*, p. 42), each affiliative post-conflict contact between former opponents or between the former victim and a third party occurring within this critical interval is considered as reconciliation and PCTA, respectively. We tested each time interval for a significant correlation between the observation period (PC vs. MC) and the cumulative frequency of affiliations using Fisher's exact test with R function fisher.test, This function also provided the respective odds ratios, of which we used the highest as a rational for determining the critical interval.

As suggested by *Aureli & van Schaik* (*1991*, p. 7), we controlled for the possibility that a difference between the cumulative distributions of PC and MC might be caused by extreme behaviour of single individuals. In order to test whether the variables *observation period (PC vs. MC)* and *initiator of affiliative contacts* were independent, we conducted a Fisher's exact test in which we included the observed frequencies of affiliative contacts within the critical time interval for reconciliation. In case the test demonstrated no significant correlation, we concluded that the found difference was not driven by extraordinary behaviour of particular individuals, but reflected a general reconciliatory tendency.

Finally, we analysed whether particular types of affiliative behaviour between the former opponents, on the one hand, and between the former victim and a third party, on the other, had been more prevalent immediately after a conflict than without a preceding conflict. We conducted Fisher's exact test and calculated odds ratios for pooled data. Additionally, we controlled for influential dyads to check whether the found effect might have been driven by the behaviour within particular dyads using a Fisher's exact test.

### Inter-rater reliability

All coding was done by KK. To assess inter-rater reliability, a second person who was a trained behavioural observer but naïve with regard to the hypothesis additionally coded 20% of the aggressive interactions and respective PCs and MCs. The calculation of Cohen's kappa coefficient ($\kappa$), using function kappa2 of R package irr (*Gamer et al., 2012*), revealed good agreement between both raters (reconciliation: $\kappa = 0.73$, $p < 0.001$; PCTA: $\kappa = 0.74$, $p < 0.001$).

## Ethical note

We observed the orangutans in their usual enclosures from the visitors' area during the zoo's opening hours. There were no manipulations of any kind or changes of their daily routine due to our study. The directorate of Dortmund Zoo provided full approval for this purely observational research. IRB approval was not necessary because no special permission for including animals in purely observational studies is required in Germany (TierSchGes §7 and §8). The Zoo Dortmund is a member of the European Association of Zoos and Aquaria (EAZA) and of the World Association of Zoos and Aquariums (WAZA). Animal husbandry and research comply with the EAZA Minimum Standards for the Accommodation and Care of Animals in Zoos and Aquaria and the WAZA Ethical Guidelines for the Conduct of Research on Animals by Zoos and Aquariums.

## RESULTS

### Aggressive interactions

In total, we recorded 114 aggressive interactions, including 16 cases of coerced copulations/copulation attempts and five counter-aggressions in the context of male sexual coercion. Whereas 85 aggressions were distinct events with no other aggression following within 4 min, the remaining 29 cases occurred in sequences of two, three or four causally connected interactions.

Half of all aggressions were of *medium intensity* ($n = 57$, 50%), followed by *high intensity* ($n = 37$, 32.5%) and *low intensity* ($n = 14$, 12.3%). In six cases (5.3%), the intensity of aggression could not be determined (for details, see Fig. S2). Even during highly intense aggression, only 16 out of 37 cases included bites or bite attempts. Only once, such an encounter resulted in a minor injury of a victim's finger.

The victim's reactions most frequently consisted of single behaviours ($n = 60$) or of combinations of up to four behaviours. The vast majority of these reactions were non-aggressive, consisting mainly in avoidance (*move away*, $n = 47$) or non-aggressive behaviours to cease the conflict (*withdraw body part/attempt to break free*, $n = 34$), frequently accompanied by vocalisations. In nine cases, the victim showed no detectable reaction. Only 20 reactions included aggressive behaviour, most of them in the context of male sexual coercion. There was no case of redirection.

The majority of aggressions seemed to occur spontaneously ($n = 49$, 43%), i.e., without any detectable reason, followed by aggressions in the contexts of sexual coercion ($n = 26$, 23%) and weaning ($n = 17$, 15%). Aggression due to food or object competition occurred only rarely in nine (8%) and six cases (5%), respectively. There was one case of third-party punishment. For the remaining cases ($n = 6$, 5%), the context was undeterminable due to limited visibility. Most high-level aggressions occurred during sexual coercion ($n = 16$, 43%), whereas most low-level aggressions occurred in the context of weaning ($n = 12$, 86%) (for details, see Fig. S2).

With regard to the different housing conditions during summer (outdoors) and winter (indoors), a one-tailed exact Wilcoxon signed rank test comparing individual aggression rates showed that they significantly increased in indoor (median = 0.144) compared to outdoor (median = 0.020) conditions ($n = 5$, $T = 0$, $p = 0.031$). The probability for highly intense aggressions to occur was 3.85 times higher indoors than outdoors (Fisher's exact test on pooled data, $p = 0.04$). This effect was not driven by the extreme behaviour of single individuals (Fisher's exact test, $p = 0.588$). Furthermore, although many indoor aggressions occurred through the mesh that separated the subgroups (illustrated as a dashed line in Fig. 1), we found no general aggression-encouraging effect of the mesh (Fisher's exact test on pooled data, $p = 1$).

Moreover, as Fig. 1 illustrates, the frequency of aggressive interactions varied *across* individuals with regard to the identity of the aggressor (indicated by the size of the vertices, i.e., circles) and *within* individuals with respect to the identity of the recipient of the aggression (indicated by the width of the respective edges, i.e., arrows). Aggressions did not occur symmetrically within dyads, but were predominantly directed down the dominance

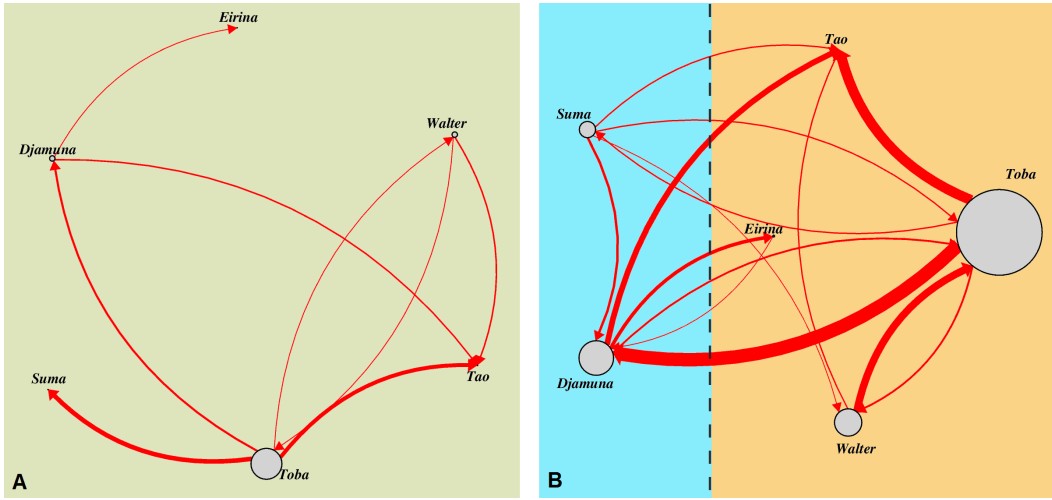

**Figure 1 Aggressive interactions for directed dyads differentiated with regard to the respective housing condition.** (A) outdoor-enclosure: all individuals were grouped together, (B) indoor-enclosures: individuals were kept in two subgroups in neighbouring enclosures (indicated by different colours) with the opportunity to interact through the separating mesh (indicated by a vertical dashed line). Vertices (circles) indicate the particular individuals with their size corresponding to the absolute number of conflicts in which the particular individual was involved as aggressor. Edges (connecting lines) indicate aggressive interactions with their size corresponding to the absolute number for the particular aggressor-victim-dyad; arrows indicate the direction of the aggression.

hierarchy. *Toba*, the highest-ranking female, performed most of the aggressions ($n = 66$, 58%), which were mainly directed towards *Djamuna* ($n = 27$), the lowest-ranking adult female. The second main target of *Toba* was her older daughter *Tao* ($n = 24$); however, these conflicts mostly emerged in the context of weaning and were usually of low intensity. Whereas *Toba* was the most frequent aggressor, she only received aggressions from *Walter* (in cases of sexual coercion), but barely from other females.

### Third-party intervention
#### All contexts except male sexual coercion
Across all contexts but male sexual coercion, one quarter ($n = 22$, 24%) of the 93 aggressions provoked an intervention by an uninvolved individual. In three of these cases, two individuals intervened, resulting in a total of 25 interventions. In one further case, it was not determinable whether an intervention had taken place or not.

The majority of interventions did not include any agonistic behaviour ($n = 17$, 68%). All individuals but *Djamuna* peacefully intervened at least once, with *Toba* and *Suma* intervening most frequently (seven and six times, respectively). Only eight interventions involved aggressive behaviour, two of them were related to the same conflict: *Toba* attacked *Djamuna* five times in support of one of her daughters; *Toba* was attacked twice by *Suma* and once by *Walter*, both supporting *Djamuna*. Individuals intervened selectively with respect to the victim's identity: e.g., *Toba* exclusively interfered in aggressions directed against her daughters, whereas *Suma* predominantly intervened when *Djamuna* had been

the target of aggression (Fig. S3). There was no case of agonistic support for the aggressor. All interventions but one by *Tao* were successful and ceased the respective aggression.

With regard to housing conditions, the probability that conflicts caused an intervention was four times greater indoors than outdoors (Fisher's exact test on pooled data: $p = 0.025$, odds ratio $= 4.24$). This effect was not driven by extreme behaviour of single individuals (Fisher's exact test: $p = 0.322$).

### The context of male sexual coercion

In contrast to other contexts, all 16 forced copulations or copulation attempts provoked at least one intervention either by *Toba*, when *Tao* was the target, or by one or both of her daughters in those cases in which *Toba* was the target. All of these interventions were physical attacks against *Walter,* including mostly a combination of behaviours such as *move between/shielding*, *grab*, *pull*, *hit* and *bite*. Whereas both interventions by *Toba* stopped the sexual aggression against *Tao*, no intervention by the immatures was successful.

## Affiliative post-conflict contacts
### All contexts except male sexual coercion

*Post-conflict affiliations between opponents.* We determined a total of 70 PC-MC pairs, of which 31 pairs were *attracted*, 23 pairs *neutral* and 16 pairs *dispersed*. The remaining 23 cases of the 93 aggressive interactions could not be included in the analysis because: (i) a further aggression occurred within 3 min ($n = 12$), (ii) a PC- or MC-contact occurred, but was not specifiable due to limited visibility ($n = 8$), or (iii) either PC or MC was not available ($n = 3$). We calculated the proportion of attracted, dispersed and neutral PC-MC pairs for each victim. An exact Wilcoxon signed rank test (with *Walter* excluded, as he had received only one aggression) demonstrated a significantly higher proportion of attracted than of dispersed PC-MC pairs ($n = 5$, $T = 0$, $p = 0.031$). Thus, according to the PC/MC method, reconciliation occurred within this study group (Fig. 2).

Applying the time rule method (*Aureli, van Schaik & van Hooff, 1989*) revealed that the majority of first affiliative contacts between the former opponents occurred within the first 1-minute interval following the aggressive interaction (Fig. 3A). A comparison of the distribution of first affiliative contacts between the former opponents demonstrated a significant difference between the PC- and the MC-condition (two sample Kolmogorov–Smirnov test: $D- = 0.545$, $p = 0.0379$). The maximum difference between the cumulative relative frequencies of PC and MC was reached after 2 minutes ($\Delta_{PC-MC} = 0.243$). In the first and second minute following a conflict, the probability for an affiliative contact to occur was 3.5 and 3.4, respectively, times higher than without a previous conflict (Fisher's exact test: $p = 0.005$ for one-minute interval; $p = 0.003$ for 2-minute interval). This probability decreased in later time intervals. This difference was not driven by extreme behaviour of single individuals (Fisher's exact test: $p = 0.967$). Following *Aureli, van Schaik & van Hooff* (*1989*, p. 42), we, therefore, operationally defined reconciliation as any affiliative contact between the former opponents within the first two minutes subsequent to the end of their conflict (Fig. 3B).

Applying this operational definition, 29 out of 70 aggressive interactions (41%) could be considered as reconciled. With regard to housing conditions, reconciliation occurred five

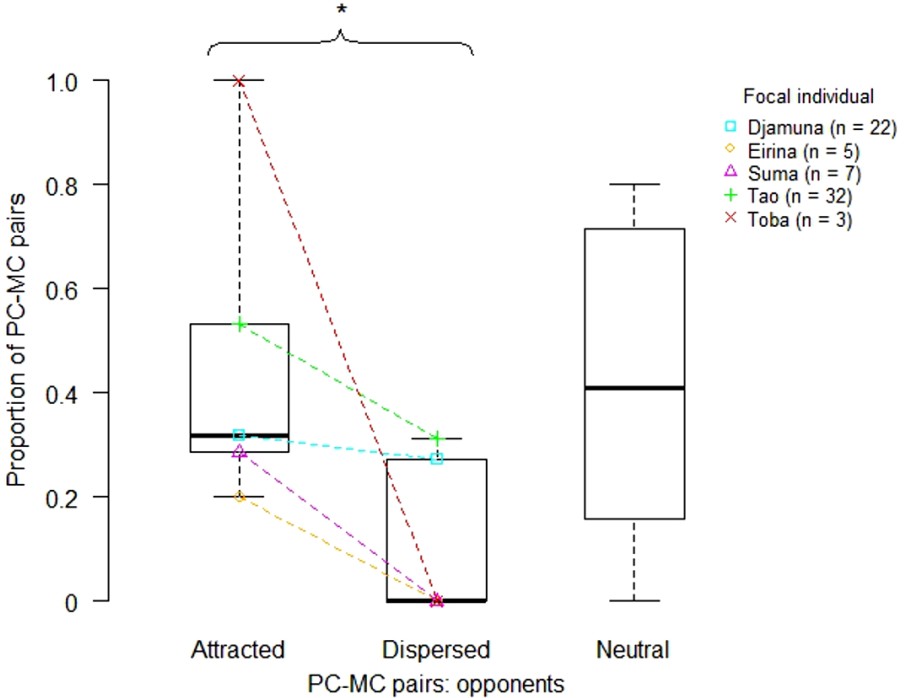

**Figure 2  Proportion of attracted, dispersed and neutral PC-MC pairs across focal individuals (victims).** Each coloured symbol indicates a particular individual. Boxplots summarise these data for attracted, dispersed and neutral PC-MC pairs with horizontal lines indicating medians, boxes indicating interquartile ranges and whiskers indicating minima and maxima. Coloured lines connect data points of attracted and dispersed PC-MC pairs for each focal individual. Numbers in brackets following the individuals' names indicate the absolute frequency of PC-MC pairs for each focal individual. Statistical significance at the $\alpha$-level of 0.05 is indicated by an asterisk above the curly bracket.

times outdoors (proportion of reconciled conflicts: 33%) and 24 times indoors (proportion of reconciled conflicts: 44%). However, a statistical test for a housing effect on reconciliation was not possible, because only three individuals provided at least three PC-MC pairs for both outdoor and indoor conditions.

Unfortunately, for the only genuine severe aggression resulting in an injury, a reunion of the opponents was impossible within 10 min. In this case, *Toba* attacked *Djamuna* through the mesh and bit her finger, whereupon *Walter* immediately forced *Toba* to copulate, which lasted about 14 min. However, 9 minutes after the aggression, *Djamuna* sat down near the mesh. Immediately after the copulation, *Toba* approached the mesh, leaned towards *Djamuna* and shaked wood wool. Within 10 min, *Djamuna* held her hand towards *Toba*, whereupon *Toba* gently touched the injured finger.

*Post-conflict third-party affiliations (PCTA).* Applying the PC/MC method for post-conflict third-party affiliative contacts resulted in 72 determinable PC-MC pairs, of which 29 pairs were attracted, 26 pairs were dispersed and 17 pairs were neutral. We calculated the proportion of attracted, dispersed and neutral PC-MC pairs for each victim and conducted an exact Wilcoxon signed rank test, which revealed no statistically significant difference
**First affiliative contacts between opponents**

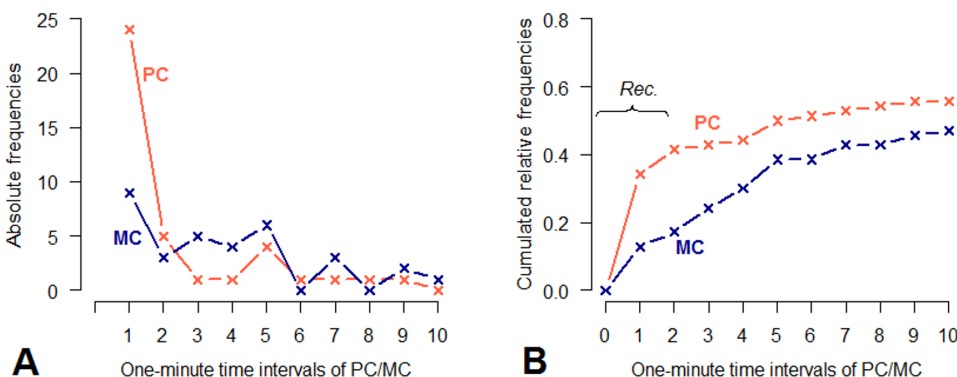

**Figure 3 Absolute frequencies and cumulative distribution over time (10 min) of first affiliative contacts between opponents in PC and MC conditions.** (A) demonstrates the number of affiliative contacts per 1-minute time interval. (B) demonstrates the cumulated relative frequencies over time, measured in 1-minute time intervals. PC observations are indicated by red, MC observations by blue lines. The bracket in graph. (B) indicates the time window in which each affiliative contact between opponents can be regarded as reconciliation (*Aureli, van Schaik & van Hooff, 1989*).

between the proportion of attracted and dispersed PC-MC pairs ($n = 5$, $T = 5$, $p = 0.313$). However, focal individuals varied with regard to their proportion of attracted and dispersed PC-MC pairs. Especially *Tao* and *Djamuna,* who were the most frequent victims, demonstrated opposing trends for attracted vs. dispersed PC-MC pairs: *Tao* 0.34 vs. 0.50 and *Djamuna* 0.43 vs. 0.22.

Applying the time rule method, we found that there were more affiliative contacts in the first minute of the PC ($n = 25$) than of the MC period ($n = 16$). However, the cumulative distribution in the PC did not significantly differ from that in the MC (two sample Kolmogorov–Smirnov test: $\hat{D}^- = 0.1$, $p = 0.9048$; Fig. 4). Thus, neither the PC/MC method nor the time rule method demonstrated a statistically significant increase of PCTA compared to control observations. However, given the small sample size and opposing trends for the two most frequently involved individuals, conclusions regarding the occurrence of conflict-induced third-party affiliations in this study group should be treated with caution.

Moreover, there was a difference with regard to the types of affiliative behaviour predominantly performed in the PC compared to the MC (Table 2). *Contact sitting* was the most frequent affiliative behaviour between victims and third parties following a conflict ($n = 24$, 45%), whereas it was much less frequent in the MC ($n = 8$, 15%). The probability for *contact sitting* to occur in PC was four times greater than in MC (Fisher's exact test on pooled data: $p = 0.002$, odds ratio: 4.0). In both conditions, *touch body* was the second-most frequent behaviour ($n = 13$, 25% for PC, $n = 14$, 26% for MC). Embrace, which has been indicated as a typical post-conflict third-party affiliation in chimpanzees (*de Waal & van Roosmalen, 1979*), occurred more often in PC ($n = 6$, 11%) than in MC ($n = 2, 4\%$); however, given the small number of instances, a statistical test was not possible.

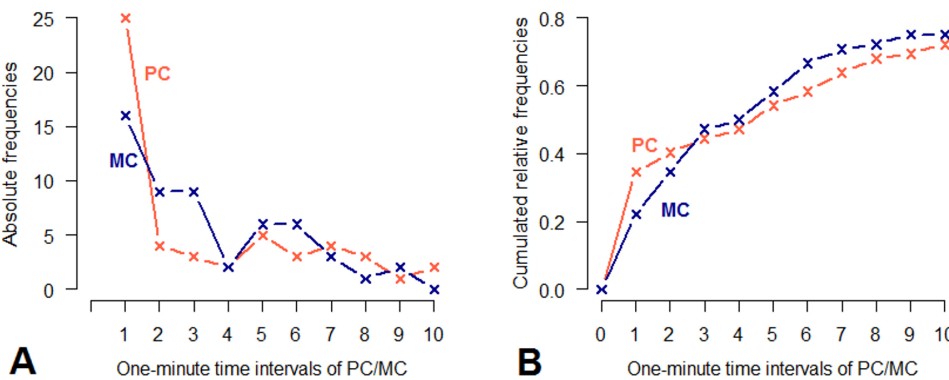

**First affiliative contacts between victims and third parties**

**Figure 4** **Absolute frequencies and cumulative distribution over time (10 min) of first affiliative contacts between victims and third parties in PC and MC conditions.** (A) demonstrates the number of affiliative contacts per 1-minute time interval. (B) demonstrates the cumulated relative frequencies over time, measured in 1-minute time intervals. PC observations are indicated by red, MC observations by blue lines.

**Table 2** **Absolute and relative frequencies of affiliative behaviours observed between victims and third parties during PC and MC, resp.** Red boxes indicate the most frequent behaviour for each condition. Proportions refer to the total number of affiliative contacts ($n = 53$ for PC, $n = 54$ for MC), they do not sum to 100% since several affiliative contacts were combinations of two behaviours.

| Affiliative behaviour | PC victim/third party | | MC victim/third party | |
|---|---|---|---|---|
| | Number | Proportion | Number | Proportion |
| *Contact sitting* | 24 | 45% | 8 | 15% |
| *Touch body* | 13 | 25% | 14 | 26% |
| *Embrace* | 6 | 11% | 2 | 4% |
| *Play* | 6 | 11% | 23 | 43% |
| *Food/object transfer* | 4 | 8% | 3 | 6% |
| *Sucking at fur/offer fur* | 3 | 6% | 7 | 13% |
| *Nursing* | 2 | 4% | 0 | 0% |
| *Grooming* | 2 | 4% | 0 | 0% |
| *Sexual interaction* | 2 | 4% | 0 | 0% |
| *Cuddling* | 1 | 2% | 3 | 6% |
| *Contact walking* | 1 | 2% | 1 | 2% |
| *Sit in close proximity* | 1 | 2% | 0 | 0% |
| *Begging* | 1 | 2% | 0 | 0% |
| *Cofeeding* | 0 | 0% | 1 | 2% |

### The context of male sexual coercion

Applying the same methods in the context of male sexual coercion, only 13 PC observations were possible, particularly because several of these aggressions followed upon each other within 3 min. The small sample size did allow neither for the application of the PC/MC method nor for the time rule method to demonstrate the occurrence of reconciliation or PCTA. However, in all cases but one, there was no affiliative contact between the former

opponents within 10 min following the conflict. On the other hand, immediately after each case of forced copulation, mutual affiliative interactions took place between *Toba* and *Eirina* while still clinging at her mother, sometimes also including *Tao*. Thus, it seems that male sexual coercion is generally not followed by a peaceful reunion of the opponents, but by PCTA between the female and her dependent or semi-dependent offspring.

## DISCUSSION

The present study systematically investigated conflict behaviour and associated conflict resolution in a group of captive Sumatran orangutans. Overall, open conflicts occurred rarely, though increased—as expected—under less spacious conditions. In accordance with our predictions, orangutans engaged in conflict resolution, especially third-party interventions and affiliative post-conflict contacts between opponents (reconciliation).

These findings support our hypothesis that the behavioural repertoire of orangutans includes prosocial conflict resolution strategies that are similar to those of the other great apes.

### Aggressive interactions

Although we applied a rather broad definition of aggression, the overall rate of aggressive interactions was low, which concurs with previous reports on captive orangutans (*Edwards & Snowdon, 1980*; *Jantschke, 1972*; *Poole, 1987*). If a narrower definition would have been used as in previous studies on third-party intervention in captive chimpanzees (*Rudolf von Rohr et al., 2012*), the aggression rate might have been substantially lower.

Whereas the specific context or cause of an aggression, e.g., sexual coercion, food competition or weaning, was often obvious to the observer, the majority of aggressive interactions between females, though, seemed to occur spontaneously, without any obvious detectable reason. Nevertheless, the *lack of identified causes* does not justify the conclusion that there *had been* no direct causes. *Judge* (*2003*, p. 60) suggested that many seemingly spontaneous conflicts might "[…] result from violations of established patterns of social interactions that human observers may not understand." For future studies, taking the relationship history of the opponents, including former conflicts, into account might help to understand some "causeless" conflicts. In any case, there will remain a considerable amount of uncertainty.

In all contexts but male sexual coercion, targets of aggression predominantly responded with avoidance or non-aggressive behaviour. Forced copulations constitute a type of sexual aggression absent in the African great apes, but common in both humans and orangutans (*Muller, Kahlenberg & Wrangham, 2009*; *Muller & Wrangham, 2009*; *Wrangham & Muller, 2009*). Our finding that forced copulations usually provoked partially fierce aggressive responses by the female is in line with observations in the wild (*Utami Atmoko et al., 2009a*). Moreover, the extent of force by the male increased with the intensity of the female's resistance, as has been reported for wild orangutans (*Knott, 2009*). Given the physical superiority of males due to the extreme sexual dimorphism in body size (*Utami Atmoko et al., 2009a*), the intensity of females' resistance is still puzzling.

## Conflict resolution
### Third-party interventions

In accordance with the few published observations of third-party interventions in captive and wild orangutans (*Marzec et al., 2016*; *Tajima & Kurotori, 2010*; *Zucker, 1987*), the majority of interventions in conflicts (except male sexual coercion) were non-aggressive and mainly performed by a dominant individual. In contrast to interventions in non-sexual contexts, which were all but one successful, all cases of forced copulation provoked immediate, aggressive, but unsuccessful interventions by one or both of the offspring, confirming findings in wild orangutans (*Utami Atmoko et al., 2009a*).

Although females seemed to intervene selectively depending on the identity of the victim, suggesting the protection of the victim, helping close social partners or kin and stabilising social bonds (*Cords, 1997*) as possible functions, there are many other potentially influential factors, particularly situational ones, which have to be considered. Likewise, the two cases of peaceful intervention by the male resembled *policing* in other primate species (*Petit & Thierry, 2000*) and seem to imply a respective function, such as maintaining dominance over group members, control over mating partners (*Boehm, 1994*; *Petit & Thierry, 2000*; *Watts, 1991*; *Watts, 1997*) or maintaining group stabilisation (*Flack et al., 2006*; *Rudolf von Rohr et al., 2012*). However, the small number of interventions does not allow for conclusions about their functions.

### Reconciliation

With this study, we were able to confirm the occurrence of reconciliation immediately after a conflict among captive orangutans. The temporal distribution of post-conflict reunions, compared to control conditions, was similar to those found for other primate species in which reconciliation occurs (e.g., *Aureli et al., 1993*; *Butovskaya & Kozintsev, 1999*; *Kutsukake & Castles, 2004*).

Several, not necessarily mutually exclusive functions for affiliative post-conflict contacts between opponents are in discussion, for example: restoring valuable social relationships jeopardised by the conflict (Valuable Relationship Hypothesis: *Aureli, van Schaik & van Hooff, 1989*; *de Waal & Aureli, 1997*), reducing uncertainty and anxiety following a conflict (Uncertainty-Reduction Hypothesis: *Aureli, 1997*; *Aureli & van Schaik, 1991*), and honestly signalling non-aggressive or benign intent to enable the resumption of non-aggressive interactions (Benign Intent Hypothesis: *Silk, 1996*; *Silk, 1997*; *Silk, 2000*).

Unfortunately, the generally rare occurrence of aggressive interactions, their unbalanced distribution across dyads and the correspondingly small and unevenly distributed number of available PC-MC pairs did not allow us to test these hypotheses. Particularly, we were not able to compute the conciliatory tendency (*de Waal & Yoshihara, 1983*; *Veenema, Das & Aureli, 1994*) in order to draw reliable conclusions about whether the orangutans reconciled selectively with particular partners. However, we consider our study as a starting point for research on conflict resolution in captive orangutans. Future studies with larger samples of captive orangutan groups should investigate the various proposed functions of reconciliation systematically. Moreover, it would be useful to also record expressions

of anxiety, such as self-directed behaviour, and to test for the effect of tension reduction through reconciliation (compare e.g., *Duboscq et al., 2014*).

Whereas we applied well-established methods to integrate our results in the large corpus of reconciliation research, these methods were also conservative. Especially the operational definition of reconciliation, by applying the time rule method, probably did not cover all affiliative post-conflict interactions that were *functionally* reconciliations. *Cords (1993)*, e.g., argued for a more functional definition of reconciliation. However, as suggested e.g., by *Veenema (2000)*, it has become common practise to combine several methods (e.g., *Koski, Koops & Sterck, 2007*; *Mallavarapu et al., 2006*; *Roseth et al., 2011*). It has also been pointed out that species-characteristics may require an adaptation of these methods (*Logan, Emery & Clayton, 2013*). We want to emphasise a further aspect, which has already been recommended for a different context (*de Waal, Leimgruber & Greenberg, 2008*): In addition to quantitative data, the qualitative description of affiliative post-conflict interactions might help to understand the proximate functions of affiliations in the aftermath of a conflict.

### Post-conflict third-party affiliations

In contrast to reconciliation, the temporal distribution of post-conflict third-party affiliations in non-sexual contexts did not differ from that of the control observations. However, though the occurrence of conflict-induced third-party affiliations with the victim could not be demonstrated with the established quantitative methods, considering qualitative observational data of post-conflict behaviour might provide further information (*de Waal & Aureli, 1997*). Following conflicts, victims and bystanders sat significantly more often in close proximity with bodily contact—often combined with *touching* or *embracing*—than without previous conflict. *Contact sitting* alongside *gentle touching* and *embracing* are typical for implicit post-conflict affiliation, especially among closely bonded partners, in several species (*Call, 1999*; *Cords, 1993*; *Fraser & Aureli, 2008*; *Verbeek, 2008*). The found higher frequency of *contact sitting* in comparison to baseline data, therefore, might hint at the occurrence of conflict-induced third-party affiliation with the victim. In contrast to findings in chimpanzees (*de Waal & van Roosmalen, 1979*), embracing did not seem to be a typical post-conflict third-party affiliation in orangutans, at least among adults. The few instances in which embracing occurred involved an immature individual. However, compared to other affiliative behavioural elements, embracing occurs generally rarely among adult orangutans (*Liebal, Pika & Tomasello, 2006*; *MacKinnon, 1974*).

The small number of forced copulations did not allow for a quantitative analysis of PCTA in this special context. Yet, immediately subsequent to each forced copulation, *Toba* and one or two of her daughters engaged in affiliative interactions. Here, consolation would be the most plausible function (*Aureli, 1997*). The stress-alleviating effect of these PCTA would probably be mutual, because not only the adult female, but also her daughters, especially the youngest one clinging at her mother during each copulation, were affected by the aggressive interactions.

Furthermore, some circumstances might have potentially confounded the detection of increased PCTA. First, given the small sample size, it was not possible to take the effect of different contexts into account (apart from differentiating between male sexual coercion

and all other contexts). A considerable proportion of conflicts occurred in the context of weaning, which—being a regular parent–offspring conflict in primates (*Maestripieri, 2002*)—is not very likely to affect other group members. The fact that the two individuals most frequently in the victim's role demonstrated opposing trends for attracted vs. dispersed PC-MC pairs, and that one of them was the semi-dependent immature involved in the weaning process, might indicate a confounding context-effect. Second, not every conflict occurred in proximity to a bystander; hence, especially low-level aggressions might have been not obvious to others. Third, although the individuals of both subgroups had the opportunity to interact with each other through the mesh—which they actually did quite often—when kept indoors, the separation potentially restricted the opportunity to initiate PCTAs for bystanders and victims from different enclosures.

As for reconciliations, the main objective of this study was to investigate whether conflicts provoke post-conflict affiliations in captive orangutans. In the case of PCTAs, this investigation did not lead to clearly positive results. Therefore, a meaningful discussion of probable functions of PCTA in orangutans is not possible at this point.

## Coping with crowded conditions

With respect to the two housing conditions, aggressions increased in both frequency and intensity under less spacious conditions. Simultaneously and in accordance to our predictions, the probability of third-party interventions increased indoors. Additionally, data of the same study group regarding food sharing demonstrated an increase in the proportion of food sharing among adult orangutan females when kept indoors (*Kopp, 2017*). These findings are in line with those of a study investigating the effect of differing housing conditions on the social behaviour of captive chimpanzees (*Nieuwenhuijsen & de Waal, 1982*). Here, the aggression rate increased indoors as did the rate of affiliative interactions. The authors interpreted this finding as a coping strategy for medium-term crowding (see also *Caperos et al., 2011*; *Judge, 2000*; *Judge & de Waal, 1997*). Moreover, whereas *Nieuwenhuijsen & de Waal (1982)* focused on affiliative behaviour in general, we specifically investigated prosocial behaviours directly associated with aggressive interactions. Therefore, the finding that bystanders ceased conflicts more likely under crowded conditions suggests that socially housed orangutans not only reduce social tension by affiliative interactions, but also flexibly use prosocial conflict resolution strategies when dispersion is not possible.

On the other hand, our results seem to be contrary to those of *Aureli & de Waal (1997)*, who found a general decrease of both affiliative and agonistic behaviour in adult chimpanzees under crowded conditions. However, the situations of the chimpanzees and the orangutans in both studies differed substantially: whereas the chimpanzees were locked up in their indoor enclosures no longer than for five consecutive days, the orangutans had already been kept indoors for about three months when we started our indoor observations. Moreover, the mean density was about ten times higher for the chimpanzees than for the orangutans in the current study. Following the authors' interpretation of the general

decrease in social behaviour as a short-term response to extremely high density (*Aureli & de Waal, 1997*), the obvious differences to our results and those of the study by *Nieuwenhuijsen & de Waal (1982)* seem to indicate different strategies to cope with varying levels of social density.

Several conflicts in the indoor enclosure occurred through the mesh that separated the two subgroups. The mesh potentially provided the opportunity to attack an individual of the other subgroup and escape an immediate retribution by withdrawing from their reach. We cannot rule out that—in particular lower ranking—individuals might have been encouraged by this opportunity, but we found no evidence for a general increasing effect on the probability of aggression.

## CONCLUSION AND FUTURE DIRECTIONS

With the present study on socially housed Sumatran orangutans, we demonstrate that the behavioural repertoire of orangutans includes a potential of conflict resolution even though this is barely needed under natural conditions.

Whereas avoidance is the predominant strategy for free-ranging orangutans to prevent conflicts, orangutans are capable to cease ongoing conflicts by third-party intervention and tend to reconcile after conflicts when living in permanent groups.

Taken into account the phylogenetic proximity of orangutans, gorillas, bonobos, chimpanzees and humans and the similarity of the respective conflict resolution strategies across these species, our findings support the hypothesis that prosocial conflict resolution is a common evolutionary heritage of all extant hominids.

In view of the dramatic loss of orangutan habitat and the increasing number of orangutans living in rescue centres, their potential to solve conflicts may become increasingly important. Future research is needed to increase our still very restricted knowledge about their abilities of conflict resolution and its functions as well as about influencing social and contextual factors. A better understanding of these crucial aspects of social life may also improve the welfare of orangutans in rescue centres and zoos due to husbandry conditions adjusted to the needs of group-living orangutans.

## ACKNOWLEDGEMENTS

We cordially thank the directorate and keepers at Zoo Dortmund for allowing us to conduct this study and their constant support in collecting the data, especially Ilona Schappert, Eddy Laudert, Natascha Kurt, Sonja Borchers and Jörg Woitzik. Special thanks go to Martin Schultze for statistical advice and to Paula Sophia Mahlke for conducting the reliability coding. Finally, we thank Jennifer Vonk as the Academic Editor, and Frans de Waal, Christine Webb and an anonymous reviewer for very helpful comments on earlier versions of the manuscript.

### Funding

This work was supported by the Freie Universität Berlin within the Excellence Initiative of the German Research Foundation. The funders had no role in study design, data collection and analysis, decision to publish, or preparation of the manuscript.

### Grant Disclosures

The following grant information was disclosed by the authors:
Freie Universität Berlin within the Excellence Initiative of the German Research Foundation.

### Competing Interests

The authors declare there are no competing interests.

### Author Contributions

- Kathrin S. Kopp conceived and designed the experiments, performed the experiments, analyzed the data, prepared figures and/or tables, authored or reviewed drafts of the paper, approved the final draft.
- Katja Liebal conceived and designed the experiments, contributed reagents/materials/-analysis tools, authored or reviewed drafts of the paper, approved the final draft.

### Animal Ethics

The following information was supplied relating to ethical approvals (i.e., approving body and any reference numbers):

The directorate at Dortmund Zoo, Germany, provided full approval for this purely observational research.

### Data Availability

The raw data are provided in a Supplemental File.

### Supplemental Information

Supplemental information for this article can be found online at http://dx.doi.org/10.7717/peerj.5303#supplemental-information.

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
