# Peer review of "Conflict resolution in socially housed Sumatran orangutans (Pongo abelii)"

_PeerJ, doi:10.7717/peerj.5303_

## Round 0.1 · original submission · Major Revisions

I have been extremely fortunate to receive such thorough reviews from three experts on this topic. All of the reviewers find merit in your study and are encouraging. I am also excited to see a paper on the social behavior of captive orangutans, as I agree that this is a long neglected area of research that can be quite illuminating with regard to the long-held notion that orangutans are relatively asocial. However, the reviewers and I share the feeling that the manuscript needs to be more tightly focused while also presenting a stronger rationale. The reviewers make some helpful suggestions for information to cut, such as the information on forced copulations. I admit I am a bit torn on this point as this is obviously useful and interesting information. I am deferring to the advice of the reviewers here though as they have more expertise than I do with this type of work.

The reviewers also provide some very helpful statistical advice that you should consider carefully.

I have a few additional comments of my own.

I feel that the introduction should delve a bit more deeply into some of the unique features of orangutan social groups in zoo settings. For example, adult males are usually separated; thus, you do not have the opportunity to witness the kinds of conflicts that are probably among the most common in wild settings. Secondly, males and females are quite dimorphic, compared to chimpanzees and bonobos, which likely factors into sex differences in response to conflicts. Here, discussion of gorilla behavior would be helpful, as one of the reviewers pointed out. The fact that there are two semi-dependent offspring with one female also needs to be treated carefully as observations involving the mother and her two offspring are not truly independent. I also agree with the reviewers who noted the need to describe in detail how the entire group could interact. It was only upon looking at Figure 2 that it was clear to me that they were all housed together when they were housed outdoors, and separated into two groups only when indoors. The observations are thus, nested within group and enclosures in a way that I am not sure you account for in your analyses.

It isn't clear what hypotheses you are testing and how you expect orangutan behavior to compare to that of other apes, or even other primates.

It wasn't clear to me how the footage was collected. If it was continuously recording, why was there only 5 hours or so of video a day? Was someone present and filming? If so, how were the recording intervals determine?

I actually like the catchy title.
In the abstract, under "background", "solitarily" should be "solitary"
Please avoid using "since" and "while" in non-temporal contexts.
signal instead of signalize on line 113.

The information in Figure 1 is better suited to a table.

·

Basic reporting

Let’s make up! – Conflict resolution in socially housed Sumatran orangutans (Pongo abelii)
Kathrin S Kopp, Katja Liebal (PeerJ – 2018)

This study is definitely of interest, because the orangutan is generally considered semi-social, or mostly solitary, so its social tendencies and capacities have been neglected. Conflict resolution is basically a potential that they don’t need much in the wild. So, why would they have it?

This question is barely addressed in the paper. One possibility is that orangutans used to be, long ago, more social, and live now (perhaps under human influence) more solitary lives. I have seen this speculation before, and would hope that the authors can discuss it in the introduction and Discussion. It would make conflict resolution a vestigial capacity.

The text often speaks about being the first to produce this study, or these data, but this kind of priority claim would be better left out. The paper is also overly long and sometimes too detailed for a scientific paper, as if a long phd thesis has been condensed into an article. I would cut even more. For example, the data on agonistic or pacific interventions don’t add much and you might consider reducing or suppressing these parts so that the paper focuses more on postconflict contacts and their function.

The sample size is relatively small (N=114), which cannot be helped, but constrains the conclusiveness of this study.

The title is a bit too popular sounding.

Specific comments

• l. 96 “mail” should be “main.”
• l. 161 aggression rate is compared by dyad, but this has the problem that dyads are not independent data points, although the stats test treats them as such. Each individual is represented in several dyads. The way this can be solved and evaluated statistically is with a random permutation analysis.
• l. 231 “proportion of agreement between both raters of 87%” – I would prefer to see a Kappa coefficient.
• l. 263 “All forced copulations but one occurred in the indoor enclosure in winter and most of them were preceded by an audible disturbance from either noisy visitors or a distressed groupmate.” – This has little to do with the topic at hand, but would be great as a separate study, especially if you could include other captive groups, because it suggests that forced copulations are not about sex but the product of stress or tension.
• Section 3.1 Do we need third party interventions in this paper?
• l. 360, it is a bit confusing to see that indoors individuals are separated by mesh yet have apparently fights and reconciliations. I have trouble seeing how this works, and it might be good to explain.
• I thought the description of the female-female reconciliation after Walter’s copulation was interesting. 

• l. 465 These interventions are described as impartial, but all I see is that they are pacific, not impartial. This is of course your point, so better explain that you are questioning the original definition.
• l. 483 “For the first time” -- Please, avoid these claims, they are annoying.
• Figures 1, 3 and 4 are unneeded.
• Figure 2, It is confusing that the various individuals are at different locations in both diagrams. It makes the comparison difficult.
• Table 2: Previous studies have compared these behavioral frequency distributions (chi-square of pooled data) to see if postconflict behavior is different from control behavior. Did you do this? I may have overlooked.

Experimental design

See above

Validity of the findings

See above

Additional comments

See above

Reviewer 2 ·

Basic reporting

The manuscript is well written. The authors report that they have demonstrated reconciliation in orangutans, a Great Ape species in which the process has not yet been shown. The lack of finding is probably partly because orangutans are semi-solitary and would not be observed under social conditions that would promote reconciliation. The authors took advantage of a zoo population living in a group to conduct the tests. As such, the results can only be generalized to this somewhat artificial zoo situation. The piece does add to the body of literature on reconciliation and the authors frame their introduction and references well. However, I have some questions concerning some of the methods, analyses and interpretation.

Experimental design

Methods
There were two subgroups. The authors should state why they were separated during the winter. In any case, the authors need to describe the flexible mesh between the two indoor enclosures in very great detail. The animals could fight between it and affiliate between it, especially since most of the PCs were recorded indoors. Would it discourage interactions in the MCs and PCs? How do two animals embrace through the mesh? Such interference would make the comparisons of types of reconciliation behavior moot. The authors state the mesh may have prevented them from seeing PCTAs (Lines 537-540), but it also may have prevented them from seeing many, many other interactions. Also, could the mesh have emboldened animals to fight if there would be no retribution? Suma had no aggression outside but aggressed against four individuals indoors. If the mesh increased the probability of fights, then all of the interpretations concerning a density effect are confounded.

The authors conduct a social network analysis (Figure 2), but do not state the particulars of how they ran it in the Data analysis section of the Methods (Lines 277-280). Also, a reader unfamiliar with SNA would not know what vertices and edges were. These should be explained.

Analyses
For the indoor versus outdoor analysis, the authors compared hourly rates of aggression for each "directed dyad" (Line 162). What is a directed dyad? Also, why use a dyadic test (Lines 162-163)? There is usually some justification as to why a test is conducted at the dyadic level rather than at the subject level (N = 6): the average rate for each subject indoors and outdoors. The Wilcoxon test (Line 276) has an "n" of 14. The authors should state where that came from. After consulting the social network analysis, I saw that 14 animals initiated aggression to others out of 30 possible pairs. This should be stated. Further, rates of aggression were reported as "average aggression rate across dyads (number of aggression per hour of observation)" (Lines 272 to 276). The authors report means but these were not what was actually compared. The authors used a nonparametric Wilcoxon test, which effectively compares medians. Perhaps medians should be reported. To directly compare means, a parametric test (e.g., t test) would need to be conducted.

In describing the PC/MC analysis, the authors state that they compared the attracted versus dispersed pairs using proportions of PCs and MCs as in Fraser and Aureli (2008). Using proportions was more appropriate for Fraser and Aureli because they did not include a subject in analyses unless it had at least three PCMC pairs. In fact, in many such studies using the PCMC method, the authors state that an animal has to have at least a certain number of PCs to be included in analysis (e.g., Castles & Whiten, 1998). Otherwise, proportions become meaningless. Walter is a good example. He only received one aggression and it resulted in an attracted pair, so the authors gave him 1.0 proportion of attracted pairs and 0.0 proportion of dispersed pairs (Figure 5). This is a very misleading analysis. There were three categories of response (attracted, dispersed and neutral). Unless Walter had at least three PCs he does not get a chance to be categorized into each category. And if Walter's one PC happened to be dispersed, the proportions would wildly swing in the opposite direction (0.0 attracted and 1.0 dispersed). Walter should be removed from a "by subject" analysis. If Walter is removed and a Wilcoxon test is conducted on the remaining proportions, the Wilcoxon test is still significant. Another option would be to run a pooled analysis of frequencies across subjects (as in de Waal & Yoshihara, 1983) and conduct a Chi-square test comparing the number of attracted to dispersed pairs against an expected ratio of 1:1. However, a pooled analysis would also be problematic because one subject (Tao) was responsible for more than half (55%) of the attracted pairs. One subject's data should not have so much influence. Further, the authors state that Toba's second most targeted victim was her daughter Tao (24 instances). The whole study might boil down to a young daughter interacting with her mother after receiving aggression from her, which the authors state was usually low-intensity aggression in a weaning context.

I recreated the frequency of attracted, dispersed and neutral pair for each subject below:

Sub Att Dis Neu Total
D 7 6 9 22
E 1 0 4 5
S 2 0 5 7
Ta 17 10 5 32
To 3 0 0 3
W 1 0 0 1
Tot 31 16 23 70

Comparing the attracted and dispersed frequencies, it does not look like an overwhelming case for reconciliation in orangutans. The median difference between attracted and dispersed pairs was 1.5 more attracted pairs than dispersed pairs. Perhaps the phrase in the title ("Let's make up!") is overstating the results a bit.

Lines 219 to 222: Please explain this test. What is an "undirected dyad"? Was it a PCMC analysis of proportion of attracted versus dispersed pairs in the first two minutes after the fight? If so, there are the same problems with proportions as mentioned above. Probably more so on a dyadic level. I still do not understand this test when it is reported in Results (Lines 349-350). Please explain in detail.

Lines 289 and 294: I do not understand the difference between the context of aggression "spontaneously" and "not determinable." If something was seemingly spontaneous, then the context cannot be determined. Was "not determinable" due to observational issues rather than context? The authors should explain the difference.

The authors start out with 114 aggressive interactions (Line 245) and end up with 70 PCMC pairs (Line 337). Apparently, coerced aggression was removed, but that leaves more than 70 fights. The authors should state how they arrived at 70 PCMC pairs.

The authors cite Aureli et al. (1989) as justification for considering all contacts within the first two minutes of aggression as reconciliation (Lines 352-354). The authors should explain this choice without the reader needing to go back and read Aureli et al. (1989). Was there a significant difference in minutes one and two? What was the criterion/rationale?

The authors report a difference in reconciliation indoors and outdoors (Lines 355-359), but do not conduct a statistical test on this difference. They should do so. If not, all of the references to a density effect on reconciliation in the Discussion is purely descriptive and should be stated as such.

In the vignette reported in Lines 360-365, the authors state that no PC was possible for this interaction. I do not understand. Djamuna was the victim. Shouldn't the authors have just focused on her for 10 minutes like all the other PCs? Certainly, Walter influenced any possible interaction between Djamuna and Toba, but this was the case between the two combatants and other animals in the group for all of the PCs. Also, the authors mention a reunion between Toba and Djamuna. How long after the initial aggression was this? Was it within the 10-minute PC period?

Validity of the findings

Interpretation
The authors imply differences in specific behavior categories in PC versus MC (Lines 387-393), but no statistical tests are conducted. Can the authors make statistical comparisons to support these statements? Again, in the Discussion, they report these differences (Lines 518-527), but these are not tested differences and descriptive observations of low numbers at best. The authors corroborate the results with other theoretical works (Lines 523-527), but the highly descriptive results do not seem justified.

The authors state "Aggressive interactions increased both in frequency and intensity under less spacious conditions" (Line 436). No tests of intensity were conducted or even described that I can see. Also concerning the density analysis, the authors state "the frequency of third party interventions to cease ongoing conflicts and reconciliation increased, both within and between the subgroups" (Lines 438-439). This was not tested either. A density effect on reconciliation was mentioned a second time (Line 447), but, again, this was not tested. The authors should acknowledge that these are not tested findings but their interpretation of very descriptive data.

Minor Points
Line 112: The observers could not know the intent of the actor "in order to cause bodily injury." Perhaps this should not be part of the definition.

Lines 314-317: The authors state that seven interventions involved aggression. When I add them up, it totals to eight.

Lines 439-441: The authors might also wish to consult Aureli and de Waal (1997) as another study of crowding in Great Apes (chimpanzees).

Additional comments

I think the data are publishable with some clarification of methods and analyses, and more circumspect interpretation of results.

·

Basic reporting

See attached .pdf.

Experimental design

See attached .pdf.

Validity of the findings

See attached .pdf.

Additional comments

See attached .pdf.

---

## Round 0.2 · Minor Revisions

All three reviewers helpfully agreed to re-review the revision of your MS and all now recommend acceptance. However, two of the reviewers have some additional minor comments that should be addressed before the paper is formally accepted. I have a few minor comments of my own:
In the background section of the abstract, I would change "...apply other conflict management strategies than avoidance.." to "apply conflict management strategies other than avoidance."

On line 38 and 43 and 247, 501, please change "while" to "Whereas". On line 166 and 396 and 469, 471, 562, please change "since" to "Because." Please be sure not to use while and since in non-temporal contexts, as indicated in my previous decision letter.
On line 130, please incorporate "Figure 1, green boxes after the citation, separated by a , rather than having adjacent parentheses.
On line 186 and 200, change & to and. Do not use & outside of parentheses.
On line 195 and 528, change "which" to "that."
It isn't clear exactly how reliability was calculated. Did you use Cohen's kappa or some other established measure of agreement?
Place a comma after i.e. on line 328.
Move the . to after the parentheses on line 342.
On line 413, change "have been" to "were."
On line 582, change "Taken these findings together, they..." to "Taken together, these findings..."

·

Basic reporting

I think the paper is greatly improved., I have no further comments.

Experimental design

N/A

Validity of the findings

See above

Additional comments

See above

Reviewer 2 ·

Basic reporting

I think the revised article now meets these standards.

Experimental design

I think the revised article now meets these standards.

Validity of the findings

I think the revised article now meets these standards.

Additional comments

The authors have addressed my comments and concerns. I think the paper is much improved by following the Reviewers’ recommendations.

I have a few minor changes and comments.
Throughout: When “third party” stands alone (e.g., “contacted a third party”), it should not have a hyphen, but when it is a two-word modifier of another word (e.g., “third-party interventions”) it should be hyphenated.

Lines numbers refer to the track changes version of the document.
Line 59: ‘has” should be “have”
Line 324: “results” should be “resulted”
Line 325: “signals” should be “signaled”
Line 981: Change “0.5” to “0.50”

Line 1018: A quote is provided but is no longer attributed to anyone.
Line 1200: A reference has been removed.

·

Basic reporting

No comment

Experimental design

No comment

Validity of the findings

No comment

Additional comments

Thank you for your thorough replies to my earlier comments, and the corresponding revisions to the manuscript. The main structural and methodological concerns I raised were addressed, resulting in a more interesting, well-framed paper. I agree with authors’ point (lines 1190-1191) - it’s an exciting starting point for orangutan conflict resolution research! My primary remaining concern: in an effort to thoughtfully consider and integrate the editor’s/reviewers’ comments, I worry the manuscript is generally a bit long and could be even more cohesive. I feel that this can be easily resolved by some paring down and editing, and am therefore happy to recommend this manuscript for publication. Just a few specific points:

- The introduction reads somewhat like a dissertation (especially the opening lines 46-49) and tends to jump around a bit. There are still occasional typos (e.g., lines 57-58, 224, 239) and the language is awkward at times (e.g., line 269: we derived the following predictions in order to test them in the present study --> we derived/tested the following predictions).
- I see that the introduction (lines 202-221) now emphasizes variation in sociability—and I think this is a nice direction. However, it’s unclear to me whether your predictions stem from the hypothesis that conflict resolution was present in the last shared ancestor of all hominids, or can be explained on a more proximate level (e.g., coping with increasingly crowded conditions in zoos/rehab centers in part due to habitat loss/fragmentation). Of course these are not mutually exclusive, but it might be nice to be more clear here, perhaps by specifying proximate/ultimate explanations.
- Lines 448-450: I still think it would be useful to indicate the unique information each approach provides—i.e., why are these the recommended methods in combination?

---

## Round 0.3 · accepted · Accept

Thank you for your patience with the mix-up with the last round of reviews. I apologize for the comments pertaining to the previous draft. I have now read the current version of the paper and am happy to accept it for publication. Your paper will make a nice contribution to the sparse literature on orangutan social behavior.

#